# Parental expectations, academic stress and non-suicidal self-injury among NEET and JEE aspirants: Examining the role of mattering as moderator

NSSI; perceived parental expectations; mattering; students; competitive exam

**Corresponding author:**
Susmita Biswas;
Email: 23dr0189@iitism.ac.in

Susmita Biswas  and Sucharita Maji

Humanities and Social Sciences, Indian Institute of Technology (Indian School of Mines) Dhanbad, India

## Abstract

Admission to engineering and medical colleges in India is determined through entrance exams, like the Joint Entrance Exam (JEE) and the National Eligibility Entrance Test (NEET), with millions of candidates facing rigorous competition. Declining mental health among these students is well documented; however, research on non-suicidal self-injury (NSSI) remains limited. The hypothesized model is based on the Integrated Theoretical Model of NSSI and Meaning in Life theory, which conceptualize NSSI as emerging from the combination of emotional vulnerability and stress, with mattering as a protective factor, respectively. A moderated-mediation model was examined. The current study aimed at testing a hypothesized moderated-mediation model to check the relationship of academic stress, perceived parental expectations, and mattering with NSSI among NEET and JEE aspirants. A total of 151 NEET and JEE aspirants ($M = 18.45$; SD = 1.97) participated in the study. A survey based on standardized questionnaires was used as a data collection tool. The reported prevalence rate of NSSI was 45.69%, which is higher than global prevalence but consistent with prior prevalence rates in Indian and South Asian samples. The study found that PPE and NSSI were indirectly associated through academic stress ($\beta = 0.128$, $b = 0.044$, SE = 0.015, $p = 0.003$, 95% CI = [0.020, 0.081]), and mattering moderated the relationship between PPE and academic stress ($\beta = -0.159$, $b = -0.063$, SE = 0.026, $p = 0.015$, 95% CI [$-0.115$, $-0.004$]). The findings suggest that mattering serves as a protective factor from NSSI in high-stakes academic environments and provides further implications for culturally informed NSSI prevention among these individuals. This further represents a novel contribution to the literature by clarifying its buffering role against NSSI among NEET and JEE aspirants.

## Impact statement

Although stressful academic conditions among regular school and college students have been studied widely, NSSI among competitive exam students who study in high academic pressure conditions in India is less researched. The current study builds upon the Integrated Theoretical Model of NSSI by exploring how academic stress (proximal determinant) and parental expectations (distal determinant) uniquely contribute to NSSI. Additionally, research based on Meaning in Life posits that mattering plays a significant role in an individual's overall mental health and suicidality. Therefore, we included mattering as a moderator in the present study to account for its conditional effects on the relationship between academic stress, parental expectations and NSSI. Results suggested that mattering had a protective effect on NSSI. Lastly, the study adds implications to policy and practice by encouraging the implementation of indicated intervention strategies such as reduction of academic stressors, teacher training and promoting a more supportive atmosphere for students in high-pressure academic environments. Finally, it also provides suggestions to educators, parents and mental health professionals with culturally sensitive interventions to help and protect such students from risky behaviors.



## Introduction

A standardized curriculum-based admission test has long served as the primary metric for measuring a student's mastery of subjects and assessing their readiness for college-level education (Atkinson and Geiser, 2009). However, admission testing has since evolved into an "educational arms race" the extreme competition to get into prestigious institutes (Atkinson, 2001). In India's competitive education system, standardized entrance exams play a decisive role in shaping students' academic careers (George, 2023). This trajectory stems from concepts such as family honor and respect, in which success confers familial and caste-based social standing, as evidenced

in the existing literature (Pal, 2025; Xu, 2025). Further, it is noted that higher student suicide rates are reported due to collectivist pressure as compared to other streams like the humanities or social sciences (Pitalia, 2024). The Joint Entrance Exam (JEE) and the National Eligibility Entrance Test (NEET) are among the most competitive exams for engineering and medical education, with millions of students competing for a limited number of seats (George, 2023). Additionally, NEET and JEE dominate the coaching industry, resulting in revenue of 6.5 million dollars, subjecting it to a profitable business model rather than solely focusing on students' careers. Thus making it ideal and timely to focus on the mental health of JEE and NEET aspirants for a deeper investigation.

Entrance tests have been associated with academic pressure, parental expectations, societal expectations and peer pressure (George, 2023). Consequently, leading to mental health issues, such as anxiety, depression, stress (Shrivastava and Rajan, 2018; George, 2023), substance use (Khalifah et al., 2023; Mamun et al., 2024), NSSI (Chen et al., 2021; Guérin-Marion et al., 2023) and poor sleep hygiene (Wang et al., 2016; Aishwarya & Rajilarajendran, 2026). Entrance exams thus emerge as a critical psychosocial stressor, linking vulnerability to deteriorating mental health among entrance test aspirants, further highlighting the interconnected pathways with maladaptive coping behaviors, like NSSI. However, behaviors like NSSI, despite being associated with entrance exams, remain understudied as a major outcome (Rabby et al., 2023; Mamun et al., 2024).

NSSI is the deliberate harm to one's own body tissues (cutting, scraping, burning and hair pulling) without the intention to die (Nock, 2010). It is common among adolescents, typically the age range of JEE and NEET aspirants, making them vulnerable to this risk (Moran et al., 2012). Unlike anxiety or depression, which mirror internal emotional states, NSSI is better understood as a coping strategy exercised to regulate stress and punish oneself to address failure. These regulatory functions reinforce NSSI, especially in a high-pressure, performance-oriented academic context, where self-worth is often tied to achievement (Szewczuk-Bogusławska et al., 2024). Moreover, parents view JEE and NEET as prestigious career paths where excelling would promise long-lasting financial stability (Gupta et al., 2025). As a result, many aspirants move to coaching centers and follow regimented lifestyles to live up to their parents' dreams. These students are under a lot of pressure from their families and institutions to perform well on these exams. This pressure has been correlated with increased risk for suicide, anxiety and depression (Gupta et al., 2025; Pal et al., 2025). Further, such a competitive environment makes JEE and NEET aspirants extremely vulnerable to NSSI; however, systematic research on this population remains scarce. Therefore, this study aims to address this gap by treating NSSI as the primary outcome variable and understanding this behavior among JEE and NEET aspirants. It also aims to suggest policy-level implications for educators and institutions in schools and coaching centers.

## Association among perceived parental expectations, academic stress and NSSI

Adolescents' perception of the degree and nature of expectations their parents have for them, particularly regarding academic achievement, professional career and personal growth, is defined as perceived parental expectations (PPE) (Sasikala and Karunanidhi, 2011; Narayanan and Alias, 2024). Parental expectations at an adequate level may induce growth and serve as a motivating factor to aim for higher academic achievements (Narayanan and Alias, 2024). However, a discrepancy between fulfilling parental expectations and actual achievement may predict distress among students (Sasikala and Karunanidhi, 2011). Though parental expectations are culture-specific (Sasikala and Karunanidhi, 2011), studies on Western students reported less PPE as compared to Eastern students, indicating higher PPE in collectivist cultures (Oishi and Sullivan, 2005).

In a collectivist country like India, education is considered a significant means of securing socioeconomic advancement and social recognition, which often leads parents to push their wards into pursuing high-status professions, such as engineering and medicine, overlooking their interests (Xu, 2025). Consequently, adolescents may feel obliged to prioritize parental expectations and internalize high pressure to avoid bringing shame to the family (Maity, 2025). These dynamics may intensify PPE, particularly when adolescents perceive a gap between expectations and a realistically achievable goal (Sasikala and Karunanidhi, 2011; Menon et al., 2024). Consequently, such expectations can extend beyond encouragement and may be internalized as relational obligations. This further heightens performance-related stress, as academic failure is perceived as a threat to fulfill family expectations and risk honor (Maity, 2025). Several empirical findings suggest that PPE is positively associated with academic stress (Subramani, 2019; Talha et al., 2020). A study (Aoki, 2019) emphasized that in collectivist cultures, PPE is closely associated with academic stress. Therefore, in addition to academic pressure, fulfilling parental expectations may also contribute to NSSI (Chen et al., 2021; Guérin-Marion et al., 2023; Wang et al., 2022a; Chen et al., 2023; Park, 2025). Hence, an indirect effect of academic stress on the association between PPE and NSSI is expected.

Prior studies aimed to understand the impact of PPE on NSSI (Chen et al., 2021; Guérin-Marion et al., 2023; Wang et al., 2022a), but a few limitations persist in the existing literature.

First, the effect of PPE on NSSI has been explored both in Western and Eastern cultures, but understudied among Indian adolescents. Indian culture and its orientation toward molding children's future based on parents' terms could be interpreted to have an intense effect on NSSI as compared to other cultures. Therefore, it is essential to understand these associations in depth and develop targeted interventions.

Second, when JEE and NEET aspirants are considered, there is a dearth of evidence in understanding the impact of PPE on NSSI. Additionally, existing studies have addressed the impact of these variables on NSSI directly, overlooking the underlying processes or factors that underpin the effect. To the best of our knowledge, no study has yet explored the underlying mechanism of the association between PPE, academic stress and NSSI. These limitations thus impact a profound understanding of NSSI.

To test the association between the PPE-NSSI pathway, the Integrated Theoretical Model of NSSI (Nock, 2009) was used to explain the development and maintenance of NSSI. The theory states that when distal risk factors (parental/peer invalidation, invalidating environment and childhood maltreatment), stressful triggers (parental expectations, family conflict and societal demands) or both are faced, it leads to proximal vulnerabilities (academic stress, poor stress tolerance and low problem-solving skills), which in turn increase the likelihood of NSSI as a maladaptive coping mechanism. Additionally, the Meaning in Life theory (Martela and Steger, 2016) posits that feeling valued or significant (mattering) may disrupt the association between the source of stress and NSSI.

### The protective role of mattering

Building on the latter framework, mattering is understood as a construct that serves a protective function by attenuating stressors. Mattering may be defined as the need to feel significant and establish meaningful relationships with others. It has two components: the perception of feeling valued and the perception of adding value (Prilleltensky, 2020). Mattering is conceptually distinct from its adjacent constructs, such as social support, belongingness and self-esteem. The adjacent constructs conceptually focus on social acceptance and being included, whereas mattering centers on interpersonal significance, that is, a sense of being important in others' lives (Flett, 2022). This fosters resilience and adaptability in youth during times of distress (Flett, 2022). In collectivist societies, such as India, China and other Asian countries, a sense of mattering is central to a student's self-worth and well-being, especially because education is closely tied to family honor and respect within society (Cao et al., 2025). This makes mattering more than an individual-level psychological need but a culturally embedded one, directly shaping how students experience academic pressure. Empirical studies consistently show that high levels of Mattering negatively impacted academic stress (Rayle and Chung, 2007; Hill and Madigan, 2022; Flett et al., 2023). Students who feel they matter demonstrate stronger resilience and coping abilities when dealing with stressors (Flett et al., 2023). Among Chinese students, particularly, high levels of mattering were associated with greater academic resilience and improved academic performance, suggesting that mattering does not simply correlate with the well-being of students, but actively buffers the influence of academic stressors (Flett et al., 2014). Therefore, drawing on mattering as the "significance" component of Meaning in Life theory, it can be hypothesized that mattering weakens the association between PPE and academic stress.

Mattering could also be considered a crucial determinant of NSSI. Mattering, as a psychosocial construct used to understand NSSI, has not been extensively studied; however, existing studies consistently show a strong association between mattering and suicidality. A review (Flett, 2022) mentioned that not feeling mattered increased the risk of suicidality among adolescents. In another study done on 2,004 youths, low levels of mattering indirectly increased suicidal ideation mediated by mental health (Elliott et al., 2005). The relationship between mattering and suicidality may be explained by the Meaning in Life theory, as discussed previously (Martela and Steger, 2016). Further, it was noted that high Meaning in Life acted as a protective factor against NSSI by increasing one's capacity to deal with distress (Chen et al., 2023). Since mattering is one of the three components of the Meaning in Life theory, it may be hypothesized that mattering may impact NSSI as well.

Therefore, as per the Meaning in Life theory (Martela and Steger, 2016), mattering would moderate the association between PPE and academic stress, such that high levels of mattering would weaken the association. To the best of our knowledge, no empirical study has directly tested the conditional effect of mattering on PPE and academic stress.

While the study aims to understand the indirect role of academic stress between PPE and NSSI, alternative models may include a direct PPE and NSSI pathway, parallel mediators such as coping and emotional dysregulation, or mattering moderating a different pathway, such as stress and NSSI. However, cross-sectional limits directionality; therefore, this study prioritizes testing the PPE-stress pathway, with mattering's theoretical buffering role.

### The current study

Through the Integrated Theoretical Model of NSSI and the Meaning in Life theory, a moderated mediation model has been constructed (see Figure 1). The model depicts PPE as a distal risk factor that triggers the proximal risk factor, that is, academic stress, which acts as a mediator, depicting an indirect association with NSSI. To further clarify, high PPE would be associated with high academic stress, which in turn would increase the risk of NSSI. Mattering, as per the Meaning in Life theory, is theorized to moderate, that is, buffer the PPE-academic stress pathway, by reducing high academic stress and weakening the association between PPE and academic stress, further reducing NSSI risk. Mattering is positioned as a moderator because it alters the strength of the relationship between PPE and academic stress rather than transmitting the effect. This aligns with Baron and Kenny's (1986) criteria, highlighting the role of the moderator in specifying "when" effects occur. In the current study, it is hypothesized that mattering will build resilience and reduce the impact of expectations on academic stress. In addition, mattering is reported to be protective against stress, without increasing risks (Flett, 2022). To test the model, the following hypotheses were proposed:

**Hypothesis 1.** The relationship between PPE and NSSI will be indirectly associated through academic stress, such that higher PPE will be associated with high academic stress, which will further be associated with high NSSI.

**Hypothesis 2.** Mattering would moderate the association between PPE and academic stress, such that higher levels of mattering would weaken the association.

### Method

#### Participants and procedure

The dataset analyzed in the study, along with the variable codebook, detailed method and measurement file, has been anonymized and made available in the open-access repository of the OSF platform. Reviewers may access the files using the attached link https://osf.io/tnh4x/overview?view_only=871cca8775f8420e802e172b5534673e. This study was reviewed and approved by the Institute Ethics Committee (IEC) at the Indian Institute of Technology (Indian School of Mines), Dhanbad. The reference number for the study is IIT(ISM)/IEC/14/2025. The required sample size was calculated using G*power 3.1. The estimation was done to ensure sufficient statistical power. Assuming a medium effect size of 0.15, power of $(1 - \beta) = 0.80$ and $\alpha = 0.05$, the minimum sample size required was 55.

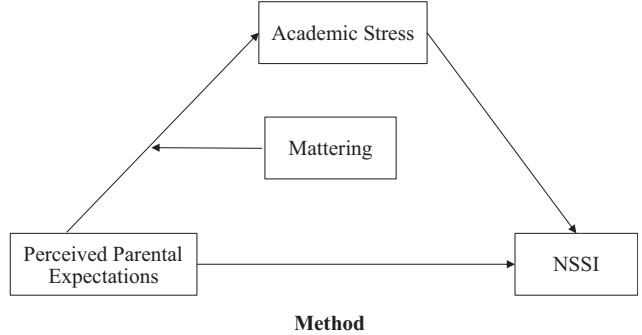

**Figure 1.** The hypothesized model of the study.

The study included 152 NEET and JEE aspirants (119 males, 33 females) residing in residential hostels in Kota, Rajasthan, India. This male dominance (78%) is consistent with prior NSSI studies on Asian populations, which reported a higher prevalence among male participants as compared to the female dominance in Western countries (Europe and North America) (Moloney et al., 2024). The participants were part of either senior secondary school level or JEE and NEET coaching centers. Participants ranged in age from 13 to 28 years (Mean age = 18.45, SD = 1.97), including both adolescents and young adults. It reflects the unique composition of JEE and NEET aspirants. Although the majority of participants are adolescents, the Indian government's removal of the upper age limit for the NEET exam makes older aspirants eligible. Additionally, the coaching culture sustains the dependence on parents both financially and emotionally, delaying the onset of youth autonomy (Pal et al., 2025). Although NSSI is typical among adolescents, older aspirants facing identical stressors are equally vulnerable to NSSI (Pal et al., 2025).

The participants were selected using a combination of purposive and snowball sampling. The data in Kota were collected in an offline mode, where all participants preparing for NEET and JEE exams at coaching centers were approached personally and invited to participate, using purposive sampling. Data were also collected in online mode, with online teaching centers contacted to reach out to these aspirants, who were requested to participate online. Those aspirants were further asked to share the questionnaire with other known aspirants, utilizing snowball sampling. While these sampling methods ensured the recruitment of participants with lived experience of NSSI, they may also have limited the sample's representativeness. Individuals with active social connections and a willingness to disclose NSSI behaviors may be overrepresented in the sample, underreporting socially isolated individuals, further influencing prevalence estimates. Additionally, employing snowball sampling increased reliance on participants' networks, introducing selection bias and further limiting the generalizability of the findings.

Initially, 248 participants were approached. Eventually, a total of 96 (38.7%) underage participants (below 18 years) were excluded as parents refused to give written parental consent. Meanwhile, 117 (47%) participants aged 18 or older provided written consent, and written parental consent was obtained from 35 (14%) underage participants. The participants were debriefed on the study. The questionnaires were presented in their original form and language (i.e., English); therefore, no linguistic adaptation procedure was used. This was deemed appropriate as the participants were proficient in English, and it is their primary medium of instruction for the sampled population. A flyer with information on mental health services was handed out to all the participants. Additionally, participants showing distressed behavior or active self-harm were offered immediate private debriefing by the lead researcher. The hostel wardens or local guardians were further informed of such situations and were given the mental health service flyer for emergencies.

## Measures

### Educational stress scale (ESS)

This 16-item self-report questionnaire measures five latent variables: Pressure from study, Workload, Worry about grades, Self-expectation and Despondency, with Cronbach's alpha of 0.878 indicating good reliability and internal consistency. It uses a 5-point Likert scale ranging from 16 to 90, with higher scores indicating greater stress (Sun et al., 2011).

In research, this scale is used to assess the impact of academic stress on adolescent mental health and well-being. This scale was originally developed for Chinese adolescents; however, it has been claimed to be suitable for use among adolescent populations from different cultures, particularly Asian cultures. The scale has also been used in multiple Indian studies to assess academic stress, with strong internal consistency ($\alpha < 0.80$), further solidifying its applicability among JEE and NEET aspirants (Pillai et al., 2023; Haritay et al., 2025). One example item of ESS is, "I feel that there is too much schoolwork." The Cronbach's alpha and Composite Reliability (CR) for the current sample were 0.89 and 0.87, respectively, indicating high internal consistency.

### Perception of parental expectations inventory (PPEI)

This is a 30-item self-report scale developed specifically for the Indian context to assess adolescents' perceptions of parental expectations. It assesses four dimensions of parental expectations, namely, Parental Expectations (PE; 10 items), Academic Expectations (AE; 8 items), Career Expectations (CE; 5 items) and Parental Ambitions (PA; 7 items; Sasikala and Karunanidhi, 2011).

This scale was used to identify the expectations of Indian parents toward JEE and NEET aspirants, specific to academics, while acknowledging the cultural constraints. In the Indian context, parental expectations are centered on academic performance and are associated with a successful career (George, 2023). Therefore, academic expectations are often experienced as a direct form of parental expectations, further justifying their use to assess parental expectations. A sample item of this scale is, "Parents expect my academic performance to make them proud." Cronbach's alpha is 0.90 for the entire scale and 0.78, 0.76, 0.65 and 0.71 for PE, AE, CE and PA, respectively, thereby showing good internal consistency (Sasikala and Karunanidhi, 2011). A 5-point Likert scale is used to score the items, with higher scores indicating greater PPE. The subscale demonstrated high internal consistency in the current sample (Cronbach's alpha = 0.86; CR = 0.86).

### General mattering scale (GMS)

This 5-item scale primarily assesses an individual's perception of how much they matter to others. The scale is scored on a 4-point Likert scale with higher scores indicating greater perceived mattering. It also exhibits good internal consistency with Cronbach's alpha value ranging from 0.63 to 0.87. A sample item of this scale is, "How important are you to others?" (Marcus, 1991).

As this scale has been positively correlated with self-esteem, positive affect and quality of life, and negatively correlated with depressive symptoms, it was used in this study to examine whether perceived sense of mattering plays a protective role against NSSI (Branquinho et al., 2024). In the current sample, Cronbach's alpha was 0.75 and CR was 0.76, indicating adequate reliability for the purpose of the study.

### Deliberate self-harm inventory (DSHI)

This scale is a 17-item behaviorally based, self-report questionnaire aimed at assessing the frequency, duration, severity and types of NSSI. A sample item of this scale is, "Severely scratched yourself, to the extent that scarring or bleeding occurred?" The Cronbach's alpha for the scale is 0.82, exhibiting a high internal consistency and good reliability. This scale is suitable to be used across adolescents

and adults. In the present sample, high Cronbach's alpha (0.85) and CR (0.87) demonstrated high internal consistency (Gratz, 2001).

## Data analysis

### Data cleaning

To ensure the accuracy and reliability of the findings of the study, data cleaning was performed. It involved assessing inconsistencies, missing values, data entry errors and outliers. On thorough investigation, no missing values or data entry errors were found. To identify multivariate outliers, Mahalanobis Distance (Ghorbani, 2019) was employed, which takes into account the correlation among variables. This approach detected one outlier among 152 data entries (participant ID: 130). The case showed a deviant response pattern; therefore, it was removed to prevent disproportionate influence on the conditional process analysis. Furthermore, removal of the outlier improved the skewness and kurtosis, ranging from $-0.556$ to $+1.967$ and from $-0.516$ to $+3.427$, respectively, indicating a reduction in extreme values and falling within the acceptable range of normality (Kim, 2013).

### Statistical analysis

The analysis began by calculating the prevalence rate of NSSI in the current sample. Second, descriptive statistics and correlation analyses were carried out using JASP v0.19.3 (JASP Team, 2024). Third, JASP v0.19.3, with the PROCESS setting, was also used to run the hypothesized moderated mediation model. Prior to the main analysis, the measurement model was tested using confirmatory factor analysis (CFA) to assess the reliability of all the scales item-wise. All variables were standardized to reduce multicollinearity associated with interaction terms and to aid the interpretation of moderating effects. The interaction terms were computed using the standardized scores of the continuous variables. The significance of effects was assessed using bias-corrected bootstrapping with 5,000 resamples; an effect was considered significant if the 95% confidence interval did not include zero (Hayes, 2014).

The model fit assessment for the measurement model was based on four goodness-of-fit measures: (a) Tucker–Lewis Index (TLI); (b) Comparative Fit Index (CFI); (c) the root mean square error of approximation (RMSEA); (d) the standardized root mean square residual (SRMR). As per the recommended cutoff criteria (Hu and Bentler, 1999), a CFI of 0.95 and above, a TLI of 0.95 and above, an RMSEA <0.06 and an SRMR of <0.08, potentially suggest a good fit for the data.

## Results

### Test of common method bias

To assess common method bias, the Variance Inflation Factor (VIF) was examined for all variables (Academic stress, PPE, etc.) involved in the study. This method ensured assessing bias across all variables. The VIF ranged from 1.14 to 1.44, which is well below the commonly accepted threshold of 3.3 (Kock, 2015). This range indicated no substantive multicollinearity. Therefore, the VIF range indicated that common method bias did not have a significant effect on the data.

### Measurement model

The three scales (ESS, PPE and GMS) were validated individually item-wise using CFA in JASP v0.19.3. The scales were loaded to the specified factors that they were supposed to measure, and the model fit was examined using Maximum Likelihood Estimation. Internal consistency was checked using Cronbach's alpha and CR, which ranged from 0.75 to 0.89 and from 0.76 to 0.87, respectively. Among all the scales, CFA removed item 8 (worry about grades domain) and item 16 (self-expectation domain) from the ESS scale due to low factor loadings (<0.40). The 14 retained items maintained representation across all domains of the scale: worry about grades (2 items), self-expectations (3 items), pressure from studies (3 items), workload (3 items), competition pressure (3 items). This comprehensive representation confirmed the revised scale's content and construct validity, capturing the core academic pressure experienced by JEE and NEET aspirants. Additionally, removing low-factor-loaded items improved the scale's model fit. The discriminant validity across constructs was also formally tested using the Fornell–Larcker Criterion (Fornell and Larcker, 1981). The method confirmed adequate discriminant validity as the square root value of AVE exceeded the inter-construct correlations. The model fit for each scale is detailed in Table 1.

### Association with demographic variables

The association between demographic variables and main variables was assessed using appropriate statistical tests in JASP. Pearson's correlation coefficient was calculated for age with all four main variables. No significant correlation was found between age and academic stress, PPE, mattering or NSSI ($p > 0.05$). Gender differences were assessed using an independent sample $t$-test. Gender was divided into two groups and coded as follows: male = 0 and female = 1. Results reported no significant differences between genders on the above-mentioned variables. One-way ANOVA was evaluated across five SES groups: 1 being the lowest SES ($n = 40$, 26.3%); 2 ($n = 31$, 20.4%); 3 ($n = 45$, 29.6%); 4 ($n = 22$, 14.5%); 5 being the highest SES ($n = 13$, 8.6%) for each of the four variables. While category 5 ($n = 13$) had limited power to detect subgroup effects, no significant differences were reported in academic stress, PPE, mattering and NSSI in the overall test.

### Descriptive statistics

In the sample of 151 participants, a total of 45.69% reported engaging in at least one incident of NSSI during their lifetime. In this study, NSSI was identified when participants endorsed one or

**Table 1.** Model fit indices with Cronbach's alpha

| Sl No. | Scale ($n$ = total items) | $\chi^2$ | df | $p$-value | Item no. Removed (below 0.8) | CFI | TLI | RMSEA | SRMR | α | CR |
|---|---|---|---|---|---|---|---|---|---|---|---|
| 1 | ESS (16) | 104 | 72 | 0.08 | Item no. 8 & 16 | 0.96 | 0.95 | 0.05 | 0.05 | 0.89 | 0.87 |
| 2 | PPE (9) | 38.63 | 22 | 0.016 | None | 0.97 | 0.95 | 0.07 | 0.05 | 0.86 | 0.86 |
| 3 | GMS (5) | 5.36 | 4 | 0.252 | None | 0.99 | 0.98 | 0.04 | 0.03 | 0.75 | 0.76 |

more items in the DSHI scale. The most frequently endorsed behavior was preventing wounds from healing ($n = 33$, 21.7%), punching oneself causing bruising ($n = 28$, 18.4%), cutting without suicidal intent ($n = 27$, 17.8%) and severe scratching with bleeding or scarring ($n = 24$, 15.8%). High-risk methods were comparatively rare, which included burning ($n = 16$, 10.5%), breaking bones ($n = 7$, 4.6%) and acid dripping ($n = 1$, 0.7%). This reflects a broad spectrum of NSSI, from moderate tissue damage to severe risky acts. Chi-square analysis was performed to examine the prevalence of NSSI across genders and socioeconomic status (SES). The analysis reported no significant differences between genders ($\chi^2 = 11.39$, $p = 0.41$) and SES ($\chi^2 = 41.19$, $p = 0.59$).

The correlation coefficients and descriptive statistics (mean, standard deviation, kurtosis and skewness) are detailed in Table 2. Academic stress, PPE and NSSI were significantly associated with one another, whereas mattering was only correlated with academic stress, not with PPE or NSSI.

### Testing for the indirect effect of academic stress

A simple mediation analysis (model 4) was conducted using PROCESS setting in JASP, to examine the indirect association of academic stress between PPE and NSSI (hypothesis 1).

PPE had a significant direct association with academic stress in Figure 2 ($\beta = 0.445$, $b = 0.629$, SE = 0.103, $p < 0.001$, 95% CI = [0.419, 0.848]), and academic stress in turn had a significant direct association on NSSI ($\beta = 0.287$, $b = 0.070$, SE = 0.021, $p < 0.001$, 95% CI = [0.034, 0.111]). However, the direct association between PPE and NSSI was not significant ($\beta = 0.128$, $b = 0.044$, SE = 0.029, $p = 0.132$, 95% CI = [0.332, 0.110]). The bias-corrected bootstrap percentile with 5,000 resamples examined the indirect association of academic stress between PPE and NSSI, which was significant ($\beta = 0.128$, $b = 0.044$, SE = 0.015, $p = 0.003$, 95% CI = [0.020, 0.081]),

**Table 2.** Descriptive statistics and correlations of the study variables

|   |   | 1 | 2 | 3 | 4 |
|---|---|---|---|---|---|
| 1 | ESS | – | | | |
| 2 | PPE | 0.445*** | – | | |
| 3 | GMS | −0.210** | 0.047 | – | |
| 4 | NSSI | 0.344*** | 0.255** | −0.127 | – |
|   | Mean | 49.377 | 29.841 | 11.907 | 1.616 |
|   | SD | 10.807 | 7.638 | 3.593 | 2.633 |
|   | Skewness | −0.122 | −0.556 | 0.181 | 1.967 |
|   | Kurtosis | −0.315 | −0.126 | −0.516 | 3.427 |

*Note:* *$p < 0.05$; **$p < 0.01$; ***$p < 0.001$.

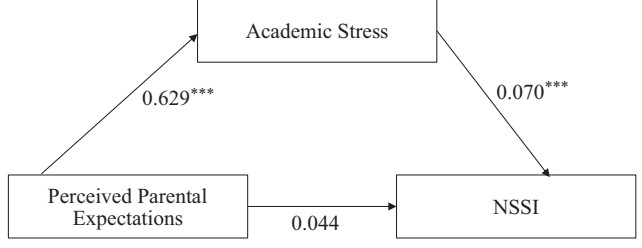

**Figure 2.** Mediation model: The indirect association between PPE and NSSI through academic stress. *Note:* ***$p < 0.001$, **$p < 0.01$, *$p < 0.05$.

with the indirect association accounting for a variance of 13.1%. Therefore, the relationship between PPE and NSSI was indirectly associated with academic stress, consequently supporting hypothesis 1, see Table 3.

### Testing for the moderated impact of mattering

The moderated-mediation model (PROCESS model 7) is visualized in Figure 3. As per the data shown in Table 4 shows that, PPE had a significant direct association with academic stress ($\beta = 0.972$, $b = 1.375$, SE = 0.317, $p < 0.001$, 95% CI = [0.569, 1.984]), which in turn is directly associated with NSSI ($\beta = 0.287$, $b = 0.070$, SE = 0.021, $p < 0.001$, 95% CI = [0.034, 0.110]). However, PPE does not have a significant association with NSSI ($\beta = 0.128$, $b = 0.044$, SE = 0.029, $p = 0.132$, 95% CI = [$-3.666 \times 10^{-4}$, 0.105]). The conditional effect of Mattering is negatively associated with academic stress, which, further, weakens NSSI ($\beta = -0.159$, $b = -0.063$, SE = 0.026, $p = 0.015$, 95% CI [$-0.115$, $-0.004$]).

The bias-corrected bootstrap method indicated that the conditional indirect association of mattering on academic stress was significant. For instance, when the level of mattering was lower (i.e., at $-1$ SD lower than the mean), the indirect association of academic stress between PPE and NSSI was positively significant and higher ($\beta = 0.181$, $b = 0.061$, SE = 0.020, $p = 0.003$, 95% CI [0.029, 0.108]). Whereas, when the level of mattering was high (i.e., at $+1$ SD higher than the mean), the indirect association was significant but weak ($\beta = 0.073$, $b = 0.026$, SE = 0.013, $p = 0.045$, 95% CI [0.006, 0.058]).

A simple slope test (Aiken, 1991; Gallucci, 2019; R Core Team, 2024; The jamovi project, 2024) was further conducted in Jamovi v2.6.44 to examine the conditional effect of mattering in the association between PPE and academic stress (see Figure 4). The association was significant at all levels of mattering, such that when mattering is low, the association between PPE and academic stress is strong, thereby increasing the likelihood of NSSI ($\beta_{\text{simple}} = 0.604$, $b = 0.854$, SE = 0.132, $p < 0.001$, 95% CI [0.593, 1.115]). Additionally, when the level of mattering was stronger, the association between PPE and academic stress was weak ($\beta_{\text{simple}} = 0.285$, $b = 0.403$, SE = 0.142, $p = 0.005$, 95% CI [0.124, 0.683]). This interaction effect supported hypothesis 2. Although mattering showed a non-significant zero-order correlation with PPE and NSSI, it emerged as a significant moderator in the conditional process analysis, in alternative pathways (PPE-academic stress). This pattern is consistent with the conditional process model, indicating that variables may have non-significant associations but be significant conditionally (Hayes, 2014). In other words, the pattern in the current study indicated that mattering did not directly influence NSSI, but did buffer against other pathways (PPE-academic stress), thereby altering NSSI behavior.

### Discussion

The current study aimed to address the gap in understanding the psychosocial factors that affect NSSI among NEET and JEE aspirants. The findings should be interpreted as per "mediation myth" precautions (Kline, 2015), which encourage considering correlational evidence as suggestive of significant associations rather than causal inference in a non-longitudinal study. The study reported that academic stress is one of the most significant factors associated with NSSI among NEET and JEE aspirants. It has a conditional indirect association between PPE and NSSI. One of the study's key

**Table 3.** Findings based on the indirect association of academic stress between PPE and NSSI

| Path | β | b | SE | p | 95% CI |
|---|---|---|---|---|---|
| PPE-academic stress | 0.445 | 0.629 | 0.103 | $p < 0.001$ | [0.419, 0.848] |
| Academic stress-NSSI | 0.287 | 0.070 | 0.021 | $p < 0.001$ | [0.034, 0.111] |
| PPE-NSSI | 0.128 | 0.044 | 0.029 | $p = 0.132$ | [0.332, 0.110] |
| Indirect association (via academic stress) | 0.128 | 0.044 | 0.015 | $p = 0.003$ | [0.020, 0.081] |

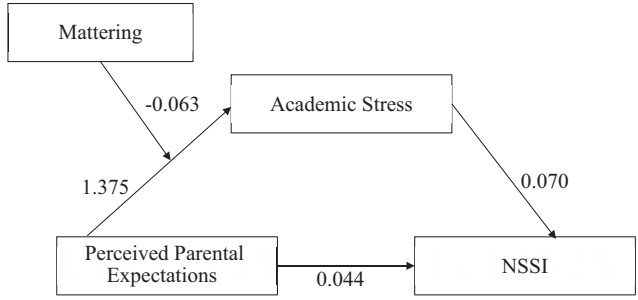

**Figure 3.** Moderated mediation model-mattering moderated the conditional association between PPE and academic stress.

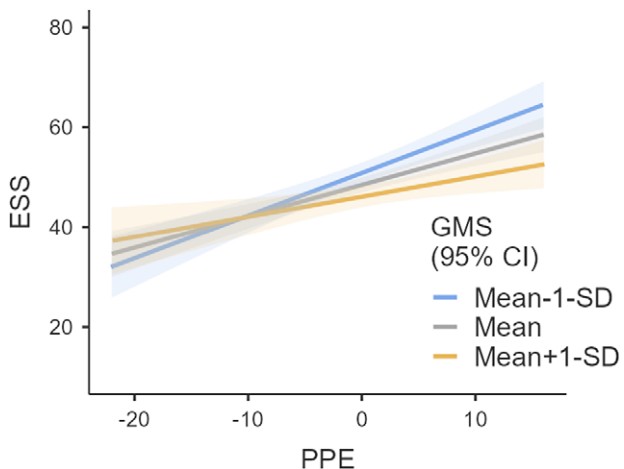

**Figure 4.** Simple slope test model indicating the buffering role of mattering in the association between PPE and academic stress.

findings was that a heightened sense of mattering buffered the conditional association between PPE and academic stress, thereby reducing NSSI engagement among the studied sample.

The study found a lifetime prevalence of NSSI among 45.69% JEE and NEET aspirants, with behaviors like preventing wounds from healing, punching oneself, cutting and severe scratching to be the most common. The observed prevalence exceeds the global prevalence of NSSI among adolescents (Wang et al., 2022a,b; Denton and Álvarez, 2024), but aligns well with the Indian and South Asian population that reported up to a 33% lifetime prevalence rate among the non-clinical adolescent population (Bhola et al., 2017; Haregu et al., 2023). The reported common NSSI behaviors in the study may reflect culturally specific strategies for addressing academic stress that are normalized. These acts, such as punching or scratching, are less stigmatized in Eastern cultures than in Western cultures that focus on cutting-based NSSI (Bhola et al., 2017). Additionally, the anonymous classroom survey administration likely enhances disclosure of such behaviors compared to clinical samples, which face greater stigma and are therefore less likely to report their NSSI (Haregu et al., 2023).

Academic stress was found to statistically explain the indirect association between PPE and NSSI, further supporting hypothesis

1. This indirect association between PPE and NSSI aligns well with the Integrated Theoretical model of NSSI (Nock, 2009). The theory states that when stressful life events/situations (high PPE) are triggered, for instance, by not doing well in academics, it increases NSSI (Chen et al., 2021). In the Indian context, where academic achievements are linked to family status (Menon et al., 2024), understanding these dynamics is extremely important to develop targeted interventions.

The study also aimed to explore the buffering of mattering on NSSI via the indirect pathway of academic stress. Provided the lack of evidence on the impact of mattering on NSSI, this study explored mattering's protective role among JEE and NEET aspirants, understanding its conditional impact on NSSI. The existing literature indicates that mattering is protective against mental health issues in adolescents, with higher mattering associated with better psychosocial well-being (Marshall and Tilton-Weaver, 2019; Flett, 2022). It also highlights that mattering significantly reduces the risk of

**Table 4.** The moderated-mediating effect of PPE on NSSI

| | Academic stress | | | | NSSI | | | |
|---|---|---|---|---|---|---|---|---|
| | β | b | t | 95% CI | β | b | t | 95% CI |
| PPE | 0.972 | 1.375*** | 4.34 | 0.569, 1.984 | 0.128 | 0.044 | 1.52 | $-3.666 \times 10^{-4}$, 0.105 |
| Mattering | | 1.223 | 1.50 | −0.773, 2.808 | | | | |
| PPE × Mattering | −0.159 | −0.063* | −2.42 | −0.115, −0.004 | | | | |
| Academic stress | | | | | | 0.070*** | 3.33 | 0.034–0.110 |
| R² | | 0.279 | | | | 0.131 | | |

*Note:* *$p < 0.05$, **$p < 0.01$, ***$p < 0.001$.

suicidality, especially among youth and young adults (Flett, 2022). Consistent with these empirical findings, this study reported evidence suggesting the protective role of mattering, further supporting hypothesis 2. The results suggested that aspirants who scored high on mattering showed a significant association with better strategies for dealing with parental expectations, without being overwhelmed by academic stress. Consequently, they were also less likely to engage in NSSI, thus supporting the Meaning in Life theory (Martela and Steger, 2016). The theory suggests evidence that high mattering was associated with reduced anxiety and enhanced the sense of worth to cope with risky behaviors like NSSI (Martela and Steger, 2016). This finding thus extends the theoretical framework and enriches our understanding of the buffering effect of mattering against NSSI in a high-pressure academic context.

Furthermore, as discussed earlier, mattering may be particularly significant in India, where self-worth is determined by family and loved ones (Markus and Kitayama, 1991). In a collectivist society, high-stakes examinations may disrupt this sense of self-worth, as academic success is associated with family honor and societal position (Chen et al., 2023). Under such dynamics, when a student perceives that they matter beyond successful academic outcomes, it may buffer against stress-induced NSSI, restoring their self-worth despite perceived social failure, as the study's findings suggest.

Although the study's findings supported the examined model, other unmeasured variables may have influenced the relationships among the variables explored. For instance, students with high levels of perfectionism, characterized by inflexible performance expectations and heightened fear of failure, may also perceive parental expectations more intensively, which could amplify academic stress and vulnerability to NSSI (Yosopov et al., 2024; Cheng et al., 2026). Further depressive symptoms may heighten risk, as they lower perceived mattering (Etherson et al., 2022). In addition, general anxiety among students may increase their sensitivity to stress and further make them more likely to engage in NSSI (Wang et al., 2026). The study did not explore the impact of these factors on NSSI along with the already examined variables. Therefore, future studies could explore the impact of these variables to better understand the broader psychological processes impacting NSSI among NEET and JEE aspirants.

## Limitations and future directions

There are several limitations to this study that should be acknowledged before interpreting its contents. First, the data collected were all self-reported by the participants; therefore, this study may be limited by response bias and social desirability. Future research may address this limitation by collecting data from multiple sources, such as parents, peers and teachers, to cross-validate findings.

Second, given the cross-sectional nature of the study, causal inferences cannot be made. As per the "mediation myth" (Kline, 2015), statistical mediation in cross-sectional data should not be interpreted as a causal process. Therefore, this study analyzes mediation pathways as statistical associations, rather than assuming causal inference. Future studies should study these effects longitudinally, as they may help understand the progression and severity of NSSI during different phases of exam preparation, that is, whether the severity increases as the exam date approaches. Longitudinal, experimental and quasi-experimental designs may help establish causal inferences. Future research could consider these designs studying mattering and academic stress. Additionally, attempting to study students longitudinally may help understand the influence of these factors on NSSI over time.

Third, the use of purposive and snowball sampling may have introduced selection bias and impacted prevalence estimates. Participants recruited through peer networks may share similar experiences and overrepresent those willing to reveal NSSI. Therefore, the generalizability of the findings is limited to a similar population.

Fourth, due to ethical requirements related to parental consent, a considerable number of underage participants were excluded, introducing selection bias. The systemic exclusion may have led to the underrepresentation of younger adolescents, limiting insights into the early stages of NSSI. Thus, it limits the generalizability of the findings to young adolescents.

Finally, the dominance of male participants (78%) in the study limits generalizability to female aspirants, as NSSI patterns may differ gender wise.

Additionally, this study focused exclusively on Indian aspirants; future research should conduct cross-cultural comparative analyses of students preparing for competitive exams globally, such as the GaoKao in China, the SAT in the USA, the CSAT in South Korea and the Abitur in Germany. This would help identify culturally specific risk factors and specify universal risk pathways leading to NSSI. It would also help test the study's pathways and aim to generalize the reported results. Finally, enable the design of targeted interventions that consider both global mechanisms and cultural dynamics.

## Implications

This study has both theoretical and practical implications for reducing NSSI. They are discussed as per the nature of the implications, that is, evidence-based and aspirational policy suggestions.

### Evidence-based implications

First, the results provide preliminary evidence that the Meaning in Life theory helps understand NSSI in academic contexts. It implies that mattering buffers against anxiety and improves self-worth by moderating the pathway from PPE to NSSI through academic stress. This provides preliminary evidence that the theory is applicable in understanding NSSI in academic contexts.

Second, promoting a supportive environment at home and in educational settings is essential to managing academic stress. Holding sessions for parents that encourage open communication may help them understand their children's experiences and problems. Parents should be encouraged to approach their children with an open mind, rather than focusing on academic achievements. Psychologists, mental health professionals and child well-being social workers should join hands to tailor programs that emphasize the importance of diverse career opportunities. In India, medicine and engineering are considered the most noble professions, and the competition to get into medical and engineering colleges is intense. Therefore, it is important to build interventions that acknowledge and validate other career paths. This will help reduce stress on students and help them choose career paths aligned with their interests, rather than being compelled to participate in the academic race.

Third, schools and coaching authorities can develop training programs for teachers that discuss, in detail, issues such as mattering and its impact on student wellbeing. They can also be trained to detect the signs of psychological distress.

Finally, acknowledging the students' efforts and providing unconditional support, whether from parents, teachers or peers, would foster a sense of mattering and reduce NSSI (Marshall and

Tilton-Weaver, 2019). The results also highlighted that mattering buffers against NSSI risk.

## Aspirational policy suggestions

Additionally, the elevated prevalence of NSSI highlights academic pressure as a concerning risk source, underscoring the need to design targeted policies. While structural reform of traditional exam culture is beyond the scope of this study, complementary strategies could be considered based on the findings. For instance, mandatory mental health screening in high-stress academic environments could be implemented (Lustig et al., 2023). Additionally, mattering-focused initiatives across schools, institutes and family settings, to minimize risk levels should be considered.

Therefore, interventions should be designed holistically, combining stress management and consideration of mattering. Implementing such strategies could help reduce NSSI and promote student well-being when facing intense academic pressure.

**Open peer review.** To view the open peer review materials for this article, please visit http://doi.org/10.1017/gmh.2026.10226.

**Data availability statement.** The dataset analyzed in the study, along with the variable codebook, detailed method and measurement file, has been anonymized and made available in the open-access repository of the OSF platform. Reviewers may access the files using the attached link https://osf.io/tnh4x/overview?view_only=871cca8775f8420e802e172b5534673e.

**Author contribution.** SB was involved in the conceptualization, data collection and analysis, and writing and editing the manuscript, and Dr. SM was involved in conceptualization, analysis and editing the manuscript.

**Financial support.** This research received no specific grant from any funding agency, commercial or not-for-profit sectors.

**Competing interests.** The authors declare that they have no known competing financial interests or personal relationships that could have appeared to influence the work reported in this paper.

**Ethics statement.** This study was reviewed and approved by the Institute Ethics Committee (IEC) at the Indian Institute of Technology (Indian School of Mines) Dhanbad. The reference number for the study is IIT(ISM)/IEC/14/2025. Participants' consent was obtained in writing.

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
