## [Reviewer Report]

<b>Major Comments</b>

<b>1. Conceptual Clarity and Theoretical Framing</b>

• While the study draws on two well-established theoretical frameworks, the distinction between “mattering” and related constructs (e.g., perceived social support, self-esteem, belongingness) would benefit from sharper clarification early in the introduction.

• It would strengthen the paper to explicitly justify why mattering, rather than broader protective constructs, is uniquely suited to buffer the PPE–academic stress pathway in the Indian context.

• The study would benefit from a brief conceptual diagram explanation in text (beyond Figure 1), explicitly walking the reader through how distal, proximal, mediating, and moderating processes are theorized to operate together.

<b>*Suggestion*</b>: Add a short paragraph explicitly differentiating mattering from adjacent constructs and situating it within collectivist cultural norms.

<b>2. Cross-Sectional Mediation and Causal Language</b>

• The authors appropriately cite the “mediation myth” and caution against causal inference; however, some sections (especially Discussion and Implications) occasionally imply directional effects (e.g., “mattering reduces NSSI”).

• Given the cross-sectional design, language should be consistently framed in terms of statistical association and buffering, rather than prevention or reduction.

<b>*Suggestion*</b>: Re-edit discussion and implication statements to consistently reflect correlational and conditional associations rather than causal processes.

<b>3. Sample Characteristics and Developmental Range</b>

• The wide age range (13–28 years) raises developmental concerns, as adolescence, emerging adulthood, and adulthood differ significantly in parental dependence, autonomy, and NSSI risk mechanisms.

• Although age was statistically non-significant, developmental heterogeneity remains a conceptual issue rather than a purely statistical one.

<b>*Suggestion*</b>:

• Either provide a stronger justification for treating this age group as theoretically homogeneous or

• Conduct and report sensitivity or subgroup analyses (e.g., ≤18 vs. >18 years), even if exploratory.

<b>4. Measurement Decisions and Construct Representation</b>

• The decision to use only the Academic Expectations subscale of the PPEI should be more strongly justified conceptually, given that parental ambitions and career expectations may be highly salient in JEE/NEET contexts.

• The modification of the ESS (item removal) based on CFA should be accompanied by a brief discussion of content validity implications, especially for a high-stakes academic population.

<b>*Suggestion*</b>: Clarify how narrowing PPE to academic expectations and modifying stress scale items may affect construct coverage and interpretation.

<b>5. Interpretation of High NSSI Prevalence</b>

• The reported lifetime NSSI prevalence (45.69%) is notably high and deserves deeper contextualization.

• While the DSHI captures a broad range of behaviors, readers would benefit from clarification regarding:

o Severity distribution

o Frequency thresholds

o Cultural interpretation of certain behaviors

<b>*Suggestion*</b>: Include a short paragraph contextualizing prevalence relative to Indian and international adolescent samples, and clarify what constitutes “any NSSI” in this study.

<b>6. Implications and Policy Recommendations</b>

• The implications section is thoughtful but occasionally moves beyond the evidentiary scope of the study (e.g., structural reform of national examinations).

• While such recommendations are valuable, clearer separation between evidence-based implications and aspirational policy suggestions would enhance credibility.

<b>*Suggestion*</b>: Distinguish clearly between findings-driven clinical implications and broader policy reflections.

<b>Minor Comments</b>

<b>1. Terminology Consistency</b>

• Ensure consistent use of terms such as self-harm, NSSI, and maladaptive coping strategies, avoiding interchangeability unless explicitly defined.

• Maintain consistent capitalization for scale names and theoretical models.

2. <b>Statistical Reporting</b>

• Consider reporting effect sizes (e.g., standardized coefficients) more consistently in the Results section to aid interpretation.

• Clarify whether SES categories were evenly distributed, as null findings may reflect limited power within subgroups.

<b>3. Ethics and Participant Safety</b>

• The inclusion of a mental-health services flyer is commendable.

• A brief note on how acute distress or disclosure of active self-harm was handled during data collection would further strengthen ethical transparency.

---

## [Reviewer Report]

This manuscript addresses an important and underexplored issue by examining non-suicidal self-injury among Indian students preparing for high-stakes competitive examinations. The study is theoretically well-grounded, methodologically sound, and makes a valuable contribution by highlighting the protective role of mattering within a culturally relevant framework. The authors are to be congratulated for their careful and thoughtful treatment of a sensitive topic.

Detailed section-wise suggestions for improvement are provided below.

1. Title

The title is informative and accurately reflects the key constructs and population; however, it is relatively long and dense.

a) You may consider streamlining the wording to improve readability while retaining the core elements (parental expectations, academic stress, NSSI, and mattering).

b) Ensure consistency by explicitly aligning the phrase “aspiring engineers and doctors” with the operational population of JEE/NEET aspirants used throughout the manuscript.

2. Abstract

The abstract clearly summarizes the study objectives, methods, and findings; however, greater clarity can be achieved by strengthening conceptual alignment.

a) Explicitly link the Integrated Model of NSSI and Meaning in Life theory to the moderated mediation design.

b) Briefly clarify the recruitment context (e.g., coaching institutes/residential settings) to better situate the sample.

3. Impact Statement

The impact statement is relevant and aligned with the journal’s applied focus; however, it would benefit from sharper emphasis on the study’s novelty.

a) More clearly highlight the unique contribution of examining mattering as a protective factor in the PPE–NSSI pathway among Indian competitive exam aspirants.

b) Consider reducing overlap with the discussion by focusing primarily on implications for policy and practice.

4. Introduction

The introduction provides a strong contextual foundation on competitive examinations in India and associated psychosocial stressors; however, the narrative can be tightened for greater focus.

a) Consider reducing descriptive sections and moving more efficiently toward the specific research gap.

b) More clearly differentiate NSSI from other mental health outcomes (e.g., anxiety, depression) to justify its distinct examination.

c) Strengthen the articulation of how the Indian cultural context uniquely shapes parental expectations and stress processes.

5. Theoretical Framework and Hypotheses

The theoretical grounding is appropriate and well-chosen, yet greater conceptual precision is needed.

a) Explicitly map perceived parental expectations, academic stress, mattering, and NSSI onto distal, proximal, and protective components of the Integrated Model of NSSI.

b) Clarify the exact locus of moderation by mattering (i.e., whether it buffers the PPE → academic stress path, the indirect effect, or both).

c) Consider briefly justifying the choice of mattering over related protective constructs (e.g., social support or resilience).

6. Method: Participants and Procedure

The methodology is ethically sound and generally well described; however, certain design characteristics require clearer justification.

a) The wide age range (13–28 years) introduces developmental heterogeneity and should be theoretically or analytically justified.

b) The combined use of purposive and snowball sampling should be discussed in terms of representativeness and potential bias.

c) The large exclusion of underage participants due to consent issues should be acknowledged later as a systematic limitation.

7. Measures

The selected instruments are appropriate and culturally relevant, but reporting clarity can be improved.

a) Report reliability coefficients obtained from the current sample for all scales and subscales used.

b) Provide a stronger rationale for using only the Academic Expectations subscale of the PPEI, given the multidimensional nature of parental expectations.

c) Briefly address the adequacy of the General Mattering Scale reliability in this sample.

8. Data Analysis

The analytic strategy is robust and clearly structured; however, several methodological decisions would benefit from additional justification.

a) Elaborate on the rationale for removing the identified multivariate outlier beyond statistical normality considerations.

b) Clearly specify the conditional process (PROCESS) model corresponding to the tested moderated mediation.

c) Briefly justify the standardization of variables and the approach used to assess common method bias.

9. Results

The results are clearly presented and statistically sound, yet interpretive clarity can be enhanced.

a) Explicitly distinguish between direct, indirect, and conditional effects in the narrative to aid reader comprehension.

b) Contextualize the relatively high prevalence of NSSI observed in the sample with reference to prior Indian or international findings.

c) Address the apparent discrepancy between non-significant zero-order correlations involving mattering and its significant conditional effects.

10. Discussion

The discussion is theoretically grounded and coherent; however, deeper interpretive engagement would strengthen this section.

a) Elaborate on why mattering may be particularly salient in exam-centric, collectivist academic environments.

b) Consider discussing plausible alternative explanations or unmeasured variables (e.g., perfectionism, depressive symptoms).

c) Maintain cautious language regarding mediation and causality, given the cross-sectional design.

11. Limitations and Future Directions

The limitations are appropriately acknowledged, but further emphasis would improve transparency.

a) More strongly highlight the implications of cross-sectional data, age heterogeneity, and exclusive reliance on self-report measures.

b) Expand future directions to include longitudinal, experimental, or intervention-based designs targeting mattering and academic stress.

12. Implications

The implications are thoughtful and culturally sensitive, though tighter alignment with the empirical findings is recommended.

a) Frame school-, family-, and coaching-based recommendations as actionable strategies directly informed by the model tested.

b) Position broader systemic reforms (e.g., restructuring competitive exams) as long-term policy considerations rather than direct outcomes of the current data.

---

## [Editor Report]

This manuscripts still has some major issues that require your attention before it be further considered for publication. Kindly address all the concerns highlighted by our reviewers as outlined.

---

## [Reviewer Report]

Title

• The phrase “Aspiring Engineers and Doctors” is contextually meaningful; however, it may be helpful to indicate explicitly that the sample consists of JEE and NEET aspirants in India for clarity to international readers.

Abstract

• The prevalence of NSSI (45.69%) is reported, but the abstract could briefly contextualize this figure in relation to prior Indian or global prevalence estimates.

• Causal wording should be moderated given the cross-sectional design.

Introduction

• The cultural framing of parental expectations within collectivist contexts is valuable; however, the argument would benefit from deeper engagement with Indian-specific sociocultural literature.

• Some sections are somewhat lengthy and could be streamlined to reduce repetition and improve clarity.

• The rationale for focusing exclusively on engineering and medical aspirants requires stronger justification.

Literature Review and Hypotheses

• The moderated mediation model is theoretically grounded, though the manuscript could elaborate further on why mattering is positioned specifically as a moderator rather than mediator.

• The transition from parental expectations to academic stress to NSSI is coherent, but potential alternative models could be briefly acknowledged.

Methodology

Participants and Sampling

• The use of purposive and snowball sampling limits representativeness and may influence prevalence estimates.

• The exclusion of underage participants due to parental consent refusal should be discussed more explicitly as a potential source of bias.

• The age range (13–28 years) is broad; developmental differences within this range should be acknowledged or statistically explored.

• The gender imbalance (119 males, 33 females) should be discussed in relation to NSSI research, where gender differences are well documented.

Measures

• The Educational Stress Scale was originally developed in a different cultural context; its cultural applicability in Indian aspirants should be discussed more thoroughly.

• It should be clarified in which language the instruments were administered and whether any linguistic adaptation procedures were used.

• Two ESS items were removed due to low factor loadings; the implications of this for construct validity should be addressed.

• Reporting composite reliability in addition to Cronbach’s alpha would strengthen psychometric rigor.

• The Measures section could be slightly condensed to improve readability.

Data Analysis

• It should be clarified whether discriminant validity across constructs was formally tested.

• The use of bootstrapped mediation and moderation analyses is methodologically sound.

• Given the cross-sectional design, caution is needed in interpreting directional paths.

Results

• The non-significant zero-order correlation between mattering and NSSI, contrasted with significant conditional effects, should be explained more clearly for readers unfamiliar with conditional process modeling.

• Effect sizes could be reported more consistently to aid interpretation of practical significance.

Discussion

• Cultural interpretation of parental expectations and collectivist values is insightful but could be more nuanced.

• Some policy recommendations, such as mandatory screening, may require stronger empirical justification.

• The discussion should further emphasize the limitations of cross-sectional design in establishing directionality.

Limitations

• Additional limitations could include potential self-report bias.

• The broad age range should be mentioned explicitly as a developmental limitation.

• The gender imbalance should be discussed as a limitation affecting generalizability.

Tables and Figures

• Figure captions could include slightly clearer explanations of directionality and conditional effects.

Conclusion

• With revisions addressing sampling limitations, cultural validation of measures, and stronger contextual interpretation, the manuscript would be significantly strengthened.

---

## [Editor Report]

There is need to view findings within the context of the study design (cross-sectional study). Kindly make sure you are consistent in your framing of the outcomes based on the design.

---

## [Reviewer Report]

The manuscript addresses an important and socially relevant topic in the Indian academic context and offers valuable theoretical and practical insights, particularly regarding the role of mattering. However, improvements are needed in conceptual clarity, methodological rigor, and statistical reporting to strengthen the overall scholarly contribution.

1. Title & Abstract

• The title is relevant to the study context; however, it could be made more precise by explicitly reflecting the mediated/moderated relationships examined.

• The abstract provides a general overview but lacks clear reporting of key statistical results (e.g., effect sizes, significance levels).

• The conceptual contribution (e.g., role of mattering as a buffer) is mentioned but not sufficiently highlighted as a novel contribution.

• The abstract would benefit from a more structured format: Background → Objective → Method → Results → Implications.

2. Introduction

• The problem statement can be sharpened with clear identification of the research gap.

• The linkage between constructs such as academic stress, mattering, and NSSI needs clearer conceptual sequencing.

• Some arguments appear descriptive rather than analytical; stronger theoretical grounding is required.

• The rationale for selecting mattering as a key construct should be more explicitly justified.

3. Literature Review

• The literature review covers relevant constructs but lacks systematic organization.

• Theoretical integration (e.g., Meaning in Life theory) is present but could be more deeply synthesized rather than descriptively presented.

• Several constructs are introduced, but their interrelationships are not consistently articulated.

• There is limited discussion of contradictory findings or alternative perspectives.

• More recent and high-impact studies could be incorporated to strengthen the review.

4. Conceptual Framework & Hypotheses

• The conceptual framework appears theoretically grounded but lacks visual clarity and structured presentation.

• Relationships between variables (e.g., mediation/moderation pathways) need to be explicitly and logically justified.

• Hypotheses should be stated in a clear, testable, and directional format.

• The framework would benefit from a formal path diagram instead of abstract or mixed visuals (as noted earlier in figure-related comments).

5. Methodology

5.1 Sample and Sampling

• The sample is male dominated (78%), limiting generalizability across gender

• Sampling limitations should be more critically discussed earlier in the methodology section.

• The demographic profile needs more detailed reporting (e.g., socio-economic background, urban/rural distribution).

5.2 Measures

• Although reliability is reported, construct validity (CFA, AVE, discriminant validity) is not sufficiently addressed.

• The absence of linguistic adaptation despite Indian context should be justified

• More justification is needed for cross-cultural applicability of scales.

5.3 Data Analysis

• While data cleaning is rigorous, the manuscript lacks clarity on analytical models used (e.g., PROCESS model details).

• There is insufficient explanation of assumptions testing (normality, multicollinearity, etc.).

6. Results

• The presentation lacks:

o Clear tables summarizing key findings

o Effect sizes and confidence intervals

• Mediation findings are interpreted, but causality is overstated, despite cross-sectional design

• Figures are not adequately explained or integrated with the text.

7. Discussion

• Some interpretations appear overgeneralized beyond the data.

• The discussion would benefit from clearer linkage back to hypotheses.

8. Implications

• The manuscript would benefit from distinguishing:

o Evidence-based implications

o Speculative or policy-level recommendations

---

## [Editor Report]

generally the arguments presented appear descriptive rather than analytical-there is need for stronger theoretical grounding and furthermore clarity around the rationale for selecting “mattering” as a construct is lacking.